# Gene variations and sweet taste sensitivity in Zambian adults with and without type 2 diabetes mellitus

Tuku Mwakyoma[1‡], Catherine Anna-Marie Graham[2‡], Benson M. Hamooya[3], Lweendo Muchaili[3], Memory Ngosa[3], Joreen P. Povia[4], Leta Pilic[5‡], Sepiso K. Masenga[3,4‡]*

1 Livingstone University Teaching Hospital Laboratory Department, Livingstone, Zambia, 2 Precision Nutrition, Lake Lucerne Institute (LLUI), Vitznau, Switzerland, 3 Mulungushi University, School of Medicine and Health Sciences, Livingstone, Zambia, 4 Livingstone Center for Prevention and Translational Science, Livingstone, Zambia, 5 Optimyse Nutrition LTD, United Kingdom

‡ TM and CA-MG share first authorship on this work. LP and SKM are joint senior authors on this work.
* sepisomasenga@lcpts.org

## Abstract

### Background

Sweet taste perception and preference play crucial roles in dietary habits and health outcomes. Understanding the genetic basis of taste thresholds and preferences can provide insights into individual differences in dietary behavior and susceptibility to metabolic disorders such as type 2 diabetes mellitus (T2DM). In Zambia, there is paucity of data concerning taste perception and preference in relation to genetics among diabetic and non-diabetic individuals. This study aimed to determine the relationship between the genotype and sweet taste thresholds, among individuals with and without T2DM in Zambia.

### Methods

A cross-sectional study was conducted among 89 adults at Livingstone University Teaching Hospital (42 non-diabetic and 47 diabetics). Saliva samples were used to determine the TRPV1 rs4790522, and TAS1R3 rs307355 genotype. We assessed sweet taste threshold and preference using a series of aqueous sucrose solutions. Demographic characteristics, anthropometrics, lifestyle factors, and dietary habits were collected using a structured questionnaire.

### Results

Sweet taste threshold positively correlated with preferred concentration in both groups (p < 0.05). A higher proportion of PwT2D with elevated preferred sweet concentrations carried one or both homozygous risk alleles (77.8%, TT/AA). When compared to healthy controls, PwT2D had higher BMI, systolic and diastolic blood

**Data availability statement:** The data underlying the results presented in the study have been provided as supporting information S2.

**Funding:** This work was supported by the Hand research group, Mulungushi University, St Mary's University, Twinkenham, London, the Fogarty International Center and National Institute of Diabetes and Digestive and Kidney Diseases of the National Institutes of Health grants R21TW012635 (SKM) and the American Heart Association Award Number 24IVPHA1297559 https://doi.org/10.58275/AHA.24IVPHA1297559.pc.gr.193866 (SKM). The content is solely the responsibility of the authors and does not represent the official views of the National Institutes of Health and the American Heart Association. The funders had no role in the study design, data collection and analysis, decision to publish, or preparation of the manuscript.

**Competing interests:** The authors have declared that no competing interests exist.

pressure, and pulse rate. They also exhibited higher taste thresholds but lower preferred concentrations, though this group was significantly older, potentially confounding results.

## Conclusion

These findings suggest taste perception and genetic variation may differ in PwT2D, highlighting the need for further research in Sub-Saharan African populations to inform personalized, cost-effective treatment strategies. However, studies with a larger sample size are required to validate our findings.

## Introduction

Diabetes is responsible for 1.5 million fatalities per year [1]. In Zambia, it was noted that there were 1.36% fatalities related to Diabetes, with an age-adjusted death rate of 35.64% per 100,000 people [2]. Unhealthy diets are identified as significant risk factors for type 2 diabetes mellitus (T2DM) [3]. High sugar intake was one of the top three dietary risk factors contributing to mortality among people living with T2DM (PwT2DM) [4]. According to the World Health Organization (WHO), on average, an adult is recommended to take not more than 25 grams of sugar each day [5,6].

Food consumption is influenced by personal experience, dietary habits, hormones, neurotransmitters, and genetic variations [7]. These determine the development of or prevent inadequate body weight management that may lead to diabetes [8]. Diabetes is thought to have a complex etiology, with genetic, nutritional, physical activity, and environmental variables all playing a role [9]. Dietary sugars have long been thought to raise the risk of T2DM, with a strong biological basis [10]. For example, glucose and maltose generate a quick rise in postprandial glycemia and are independent risk factors for T2DM [11].

Sweet taste sensitivity is defined as the least concentration at which the individual may perceive a given sweet attribute [12] and it may be influenced by genetic differences in the sweet taste receptors such as the taste 1 receptor member 3 (TAS1R3) and other taste receptors [13]. According to some research, persons who are less sensitive to sweet food may consume more sugary meals to compensate for their inability to taste them as intensely, this is commonly observed in prediabetes [14]. Individual variations in sweet taste perception have received less attention, with conflicting results [15]. Sweet taste sensitivity has been linked to the genetic ability to perceive prototypical flavor or stimuli [16]. Sweet flavor has also been connected with sweet food intake and the number of sweet taste receptors (G-protein coupled receptor) [15,17,18]. Excess sugar consumption has been linked to obesity, metabolic disorders, diabetes, cardiovascular disease, cancer, depression, and cognitive impairment [9,19].T2DM is a systematic condition that is particularly prone to these taste disorders [20]. Taste alterations progress with duration of diabetes [21]. PwT2DM tend to consume more sugar than those without it [22].

Taste impairment may be caused by a genetic or acquired taste receptor defect, abnormal brain mechanisms, peripheral neuropathy affecting taste nerves, or microangiopathy affecting taste buds [23]. The exact pathophysiology is unknown. Research suggests that high blood glucose levels can negatively affect taste perception, leading to a cycle of poor glycemic management [24]. Some researchers found no association.

The TAS1R3 coding sequence substantially correlates with human sucrose taste sensitivity and accounts for 16% of population diversity in perception [25]. In this study we selected to explore the variations of the TAS1R3 gene because it encodes the T1R3 receptor protein, which is involved in sweet taste perception [26]. Genetic variations in the TAS1R3 gene can lead to differences in how one perceives sweetness and umami savory tastes [27]. We also looked at the TRPV1 (transient receptor potential vanilloid 1) channel, it's mostly connected with perceiving heat and pain [28]. TRPV1 is co-expressed with taste receptors TAS1Rs and TAS2Rs and acts as a downstream component in sweet, bitter, and umami taste signal transduction [29]. Certain sweeteners, such as saccharin and aspartame, have been shown in studies to sensitize TRPV1 receptors, potentially improving sweet taste perception [30], this shows that TRPV1 may influence how sweet flavours are perceived in the mouth.

Few studies have analyzed the relation of the allelic diversity of these genes and sweet taste sensitivity among diabetes and non-diabetes in sub-Saharan Africa [25]. Current research focusses mostly on European populations, with few studies investigating genetic variables influencing sweet taste preferences in African people [15]. Recently, a genome wide association study (GWAS) focused on understanding the determinants of sweet liking in the 426 African and East Asian ancestry groups in the U.S found significant differences in allele frequencies and effects on sweet liking [31].

While the link between sweet taste sensitivity and diabetes has been examined in other populations, there is a dearth of literature coming from the Sub-Saharan Africa [24]. The scarcity of studies on sweet taste genetics in African populations reveals a huge gap in understanding how these genetic factors influence dietary patterns and health outcomes, notably diabetes. More research is needed to better understand these genetic links and their consequences for nutrition and health in Sub-Saharan Africa. This exploratory study first aimed to understand how the *TAS1R3* rs307355 and *TRPV1* rs4790522 SNP influence sweet taste detection threshold, preferred sample concentration, and other health markers, in healthy participants and in PwT2D in a Sub-Saharan African population. Exploratory comparisons were made between disease status.

## Method

### Study design and Subjects

We conducted an analytical cross-sectional study at Livingstone university teaching Hospital, among 89 adults of whom 42 and 47 were non-diabetic and diabetic participants, respectively, in September 2023. All subjects were originally from Livingstone, Zambia. They were informed of the study's purpose and protocol (verbally and in writing). The 42 healthy participants included hospital health workers and students within the premises that did not have diabetes. The 47 PwT2DM were recruited during their usual procedures at the Hospital.

Eligibility criteria included participants with or without T2DM aged ≥18years. This study excluded pregnant women due to altered taste sensitivity that occurs during pregnancy. We also excluded participants with taste disorders (ageusia) and chronic illnesses other than T2DM.

### Procedure

Body weight (Kgs) and height (m) were measured without shoes during their examination for the purpose of measuring the body mass index (BMI). BMI was calculated as an individual's weight in kilograms divided by the square of the height in meters. Demographic data (age, sex) were recorded and assessed together with physical activity, smoking, alcohol habits and whether one had T2DM or not. All participants were required to test for fasting blood sugar (FBS) to ascertain whether they had T2DM or not.

 

## Sweet taste sensitivity determination sweetness

Sweet taste thresholds were determined as follows; Participants were instructed to avoid eating and drinking (except water) before tasting. Six concentrations (solutions of sucrose in water) of each taste ranging from a "hardly distinguishable" to a "distinct" taste were prepared. Spring water was used as a negative control. The solutions were given in increasing concentrations in each taste series. The following concentrations were used: sucrose in spring water (sweet) in the concentrations 5.0, 15.0, 30.0, 60.0, 120.0 and 240.0 g/L. Participants rinsed their mouth with water for one minute and then swirled the test solutions in the mouth for approximately 10 seconds before expectorating. The sweet taste threshold, i.e., the concentration at which the respondent can distinguish the sweet taste from water, and the concentration that a participant preferred, i.e., the concentration the respondents picked as their favorite, were recorded.

## Collection of saliva samples and single nucleotide polymorphism (SNP) genotyping

Pure genomic DNA was isolated from 2 ml saliva samples provided by the participants. Saliva samples were collected by the research team in Livingstone, Zambia and stored at −20 °C on the premises of the Mulungushi University until shipped to St Mary's University, Twickenham, England where they were stored at −20 °C and subsequently analyzed. DNA extraction (PSP® SalivaGene 17 DNA Kit 1011), quantification (Nanodrop; Thermo Fisher Scientific), and genotyping (StepOnePlus thermocycler; Applied Biosystems, CA, USA) were carried out as previously reported [32]. Genotyping was performed using a predesigned TaqMan® SNP genotyping assay for rs307335, rs8065080, rs4790522. The primers and the probes were predesigned by Applied Biosystems. Two technical replicates were analysed for each sample. Genotyping call rate was > 95%. The rs8065080 minor genotype frequency was not represented in our sample and thus was excluded from further analysis. *Data Analysis Plan*

Normality was checked using Shapiro-Wilk test. Healthy participants and PwT2DM were analyzed separately, and then exploratory analysis was carried out between cohorts.

Firstly, for within analysis of healthy participants and PwT2DM, Pearson' correlation analysis was used between sweet taste and preferred sample threshold. Then genotypes in an additive, dominant, recessive or combined model were used as grouping variables (TAS1R3; Taste 1 Receptor Member 3 gene, rs307355; TT (wildtype), CT, CC, TT/CT (recessive model), CT/CC (dominant model); TRPV1; Transient Receptor Potential Cation Channel Subfamily V Member 1 gene, rs4790522; CC (wildtype), AC, AA, AC/CC (recessive model), AA/AC (dominant model); Combined No Risk; homozygous wildtype and homozygous wildtype/heterozygous groups) to detect differences between those with a high or low taste threshold and preferred sample concentration (Low Threshold; ≤ 15 g/L, High Threshold; > 15 g/L, Low Preferred Concentration; ≤ 30 g/L, High Threshold; > 30 g/L). Chi Squared or Fishers Exact tests were used where appropriate, data is presented with actual and expected frequencies.

Following this, BMI, SBP, DBP, and pulse were assessed in the same manner regarding genotype groupings. One way ANOVA, T-test, Kruskal Wallis, or Mann Whitney U tests were used where appropriate.

As exploratory analysis age, sex, BMI, SBP, DBP, pulse, taste threshold and preferred sample concentration were compared between disease status. T-test, Mann Whitney U, Chi Squared or Fishers Exact were used where appropriate. Then, genotype grouping frequencies were compared using Chi Squared or Fishers Exact tests where appropriate. P-value significance was set at 0.05, and SPSS (IBM SPSS Statistics, Version 30) was used throughout, with figures created in GraphPad (PRISM, version 10.4.3).

## Ethical approval

Ethical clearance was sort from Mulungushi University, School of Medicine and Health Sciences Research Ethics Committee (MUHSREC) (Assurance No. FWA0002888 IRB00012281 of IORG0010344) on the 3rd of May 2023 and the National Health Research Authority (NHRA) on the 23rd of August 2023. Permission from Livingstone central Hospital was granted. We provided consent forms to our participants and explained to them, that their participation was voluntary and that they

were free to withdraw at any time, without giving a reason and without cost. Written Consent was obtained from our participants by signing the consent form. There was no personal identifying data acquired; therefore, participants could not be identified.

We used the Strengthening the Reporting of Observational Studies in Epidemiology to guide the reporting (S1 STROBE).

## Results

### Healthy participants

Forty-two healthy participants were analysed aged 28.5±13.3 years (median±IQR)), with a mean BMI of 24.5±4.9 (Table 1). All genotypes were assessed in an additive, recessive and dominant model throughout.

Sweet threshold and preferred sample concentration were positively correlated (r=0.53, p<0.001; Fig 1). Sweet threshold and preferred sample groupings (High/Low) did not differ based on rs307355 nor rs4790522 genotypes (Fig 2).

No differences were found between rs307355 nor rs4790522 genotypes in BMI, SBP, DBP, or pulse (Table 1).

Lastly, when *TAS1R3* rs307355 and *TRPV1* rs4790522 genotype were considered together, no differences were found in BMI, SBP, DBP, or pulse (S1 Table 1). Nor with threshold or preferred concentration (p>0.05; Fig 3).

**Type 2 diabetes mellitus.** Forty-seven PwT2D were analysed aged 55.9±13.0 years (median±IQR)), with a mean BMI of 27.9±5.5 (Table 2). All genotypes were assessed in an additive, recessive and dominant model throughout.

Sweet threshold and preferred sample were positively correlated (r=0.64, p<0.001; Fig 1). Sweet threshold and preferred sample grouping did not differ based on genotype (p>0.05; Fig 4).

No differences were found between rs307355 nor rs4790522 genotypes in BMI, SBP, DBP, or pulse (p>0.05; Table 2).

Lastly, when *TAS1R3* rs307355 and *TRPV1* rs4790522 genotype were considered together, no differences were found in BMI, SBP, DBP, or pulse (S1 Table 1). However, more PwT2D that had a high preferred concentration carried one or

**Table 1. Healthy participant characteristics (mean±SD), in the total cohort and by *TAS1R3* rs307355 and *TRPV1* rs4790522 genotypes, in an additive, recessive and dominant manner.**

| TAS1R3 | Total | +TT | CT | *CC | p-value | *TT/CT | p-value | +CT/CC | p-value |
|---|---|---|---|---|---|---|---|---|---|
| No. | 42 | 9 | 20 | 13 | | 29 | | 33 | |
| Age (yrs) | 31.2±9.9 | | | | | | | | |
| Sex (%F) | 50% | 22% | 70% | 38% | | 55% | | 58% | |
| BMI (kg/m²) | 24.5±4.9 | 23.5±4.8 | 25.3±5.8 | 24.0±3.3 | 0.588 | 24.8±5.5 | 0.652 | 24.8±5.5 | 0.695 |
| SBP (mmHg) | 114.7±14.0 | 115.8±14.0 | 112.1±12.4 | 117.8±16.4 | 0.639 | 113.2±12.8 | 0.329 | 114.4±12.8 | 0.608 |
| DBP (mmHg) | 70.3±10.1 | 68.4±9.8 | 69.4±8.9 | 72.8±12.1 | 0.617 | 69.1±9.0 | 0.136 | 70.8±9.0 | 0.548 |
| Pulse (bpm) | 71.9±12.8 | 69.6±11.9 | 73.2±12.1 | 71.5±15.1 | 0.787 | 72.0±12.0 | 0.144 | 72.5±12.0 | 0.547 |
| TRPV1 | Total | +CC | AC | *AA | p-value | *AC/CC | p-value | +AA/AC | p-value |
| No. | 41 | 12 | 20 | 9 | | 32 | | 29 | 42 |
| Age (yrs) | 31.0±9.9 | | | | | | | | |
| Sex (%F) | 50% | 50% | 50% | 67% | | 19% | | 52% | |
| BMI (kg/m²) | 24.4±4.9 | 23.1±5.1 | 25.3±4.7 | 23.9±5.0 | 0.436 | 24.4±4.9 | 0.744 | 24.9±4.9 | 0.238 |
| SBP (mmHg) | 114.4±14.0 | 113.1±17.6 | 117.6±12.6 | 109.0±10.9 | 0.178 | 114.4±14.0 | 0.195 | 114.9±14.0 | 0.419 |
| DBP (mmHg) | 70.1±10.1 | 66.4±12.2 | 72.5±9.5 | 69.6±7.9 | 0.171 | 70.1±10.1 | 0.865 | 71.6±10.1 | 0.083 |
| Pulse (bpm) | 72.3±12.7 | 69.2±14.7 | 73.2±11.4 | 74.6±13.4 | 0.589 | 72.3±12.8 | 0.552 | 73.6±12.8 | 0.369 |

BMI; body mass index, DBP; diastolic blood pressure, No.; participant number, SBP, systolic blood pressure, SD; standard deviation, TAS1R3; Taste 1 Receptor Member 3 gene, rs307355; TT (wildtype), CT, CC, TT/CT (recessive model), CT/CC (dominant model); TRPV1; Transient Receptor Potential Cation Channel Subfamily V Member 1 gene, rs4790522; CC (wildtype), AC, AA, AC/CC (recessive model), AC/CC (dominant model), *+ indicate comparisons made within genotypes. P-value significance at 0.05, significant values are in bold. One way ANOVA, T-test, Kruskal Wallis, or Mann Whitney U tests were used where appropriate. Non-parametric variables are indicated with ▪ and median and interquartile range is presented.

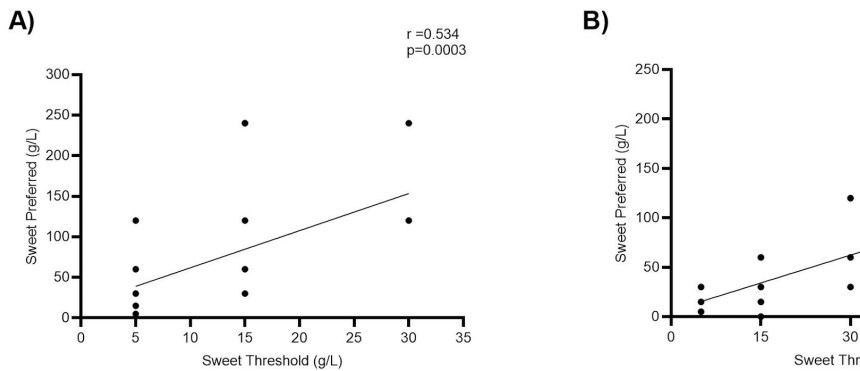

**Fig 1. Correlation between sweet taste threshold and preferred sweet taste concentration in A) healthy participants and B) participants with Type 2 Diabetes Mellitus.** *P-value significance at 0.05; significant p-vales are provided. Pearson's correlation was used throughout.*

both homozygous risk alleles (77.8%;n = 21; TT/ AA), when compared to those who with a low preferred concentration (35.3%; n = 13 p = 0.022; Fig 5). No differences were found regarding threshold (p > 0.05; Fig 5).

**Healthy participants versus participants with type 2 diabetes mellitus.** Exploratory analysis demonstrated that age, BMI, SBP, DBP, Pulse, taste threshold and preferred sample concentration differed between cohorts (p < 0.05; Table 3; Fig 6).

No significant differences were found between healthy status and genotypes (p > 0.05; S1 Table 2).

## Discussion

This study investigated whether the TAS1R3 rs307355 and TRPV1 rs4790522 SNPs influence sweet taste detection threshold, preferred sucrose concentration, and health markers in healthy individuals and PwT2D in a Sub-Saharan African population. In both groups, sweet taste threshold positively correlated with preferred concentration. Among PwT2D, a higher proportion of those with elevated preferred concentrations carried one or both homozygous risk alleles (77.8%, TT/ AA), a pattern not observed in healthy participants. Compared to healthy controls, PwT2D had higher BMI, systolic and diastolic blood pressure, and pulse rate. They also exhibited higher taste thresholds but lower preferred concentrations, though this group was significantly older, potentially confounding results. These findings suggest taste perception and genetic variation may differ in PwT2D, highlighting the need for further research in Sub-Saharan African populations to inform personalized, cost-effective treatment strategies.

Our finding regarding a lower sweet taste sensitivity in T2DM as compared to healthy participants is consistent with findings of previous studies conducted [24,33]. This diminished sensitivity to sweetness may contribute to altered dietary preferences and behaviors among individuals with diabetes, potentially influencing their overall glycemic control and metabolic health [23,33–35]. The T2DM cohort was significantly older than the healthy cohort, so results regarding comparative taste decline should be interpreted with caution. However, it is known that taste perception can decline with age [36] and diabetes risk increase with age. Therefore, healthcare providers involved in the management of T2DM should therefore consider this altered taste perception as it has potential to negate planned dietary modification plans and make effort to educate these patients on its further implications on diabetes outcomes. Increased education and awareness among diabetics on taste perception alteration and need for well-planned dietary modifications can improve their dietary choices, hence improving their overall health outcomes [37].

In this study, a greater percentage of PwT2DM preferred a lower concentration of sucrose when compared to healthy participants. This preference for lower quantities of sucrose, was also found in an investigation conducted in Korea [38]. The majority of our PwT2DM admitted to being on a diet (data not shown), however the primary cause of the lower

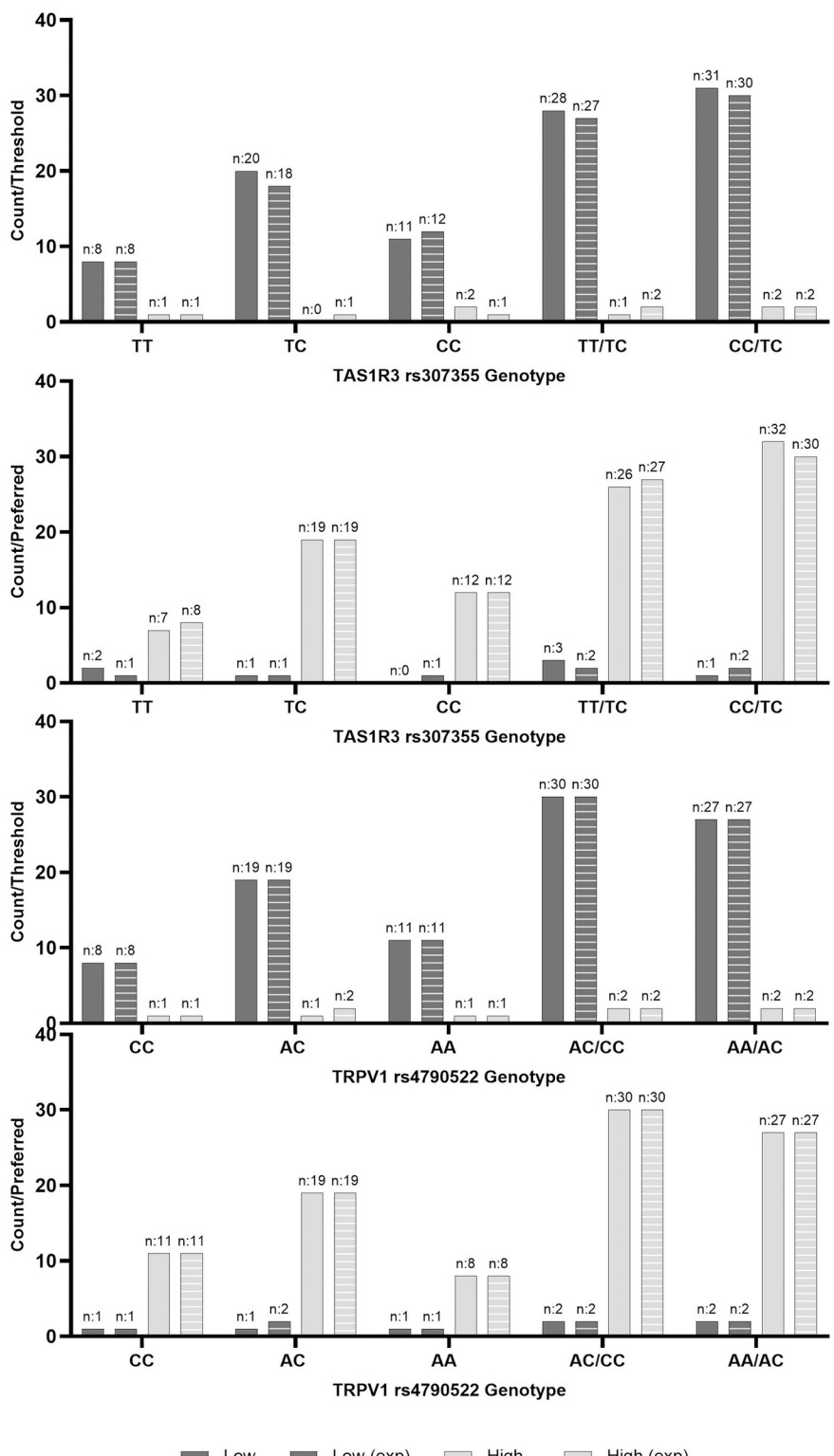

**Fig 2. Actual and expected frequency by genotype of high and low threshold and preferred concentration groupings in healthy participants.**
*Low Threshold; ≤15g/L, High Threshold; >15g/L, Low Preferred Concentration; ≤30g/L, High Threshold; >30g/L, TAS1R3; Taste 1 Receptor Member*

*3 gene, rs307355; TT (wildtype), CT, CC, TT/CT (recessive model), CT/CC (dominant model); TRPV1; Transient Receptor Potential Cation Channel Subfamily V Member 1 gene, rs4790522; CC (wildtype), AC, AA, AC/CC (recessive model), AC/CC (dominant model). P-value significance at 0.05, no significance found. Figure represents Chi Squared contingency actual and expected counts between additive, recessive and dominant models.*

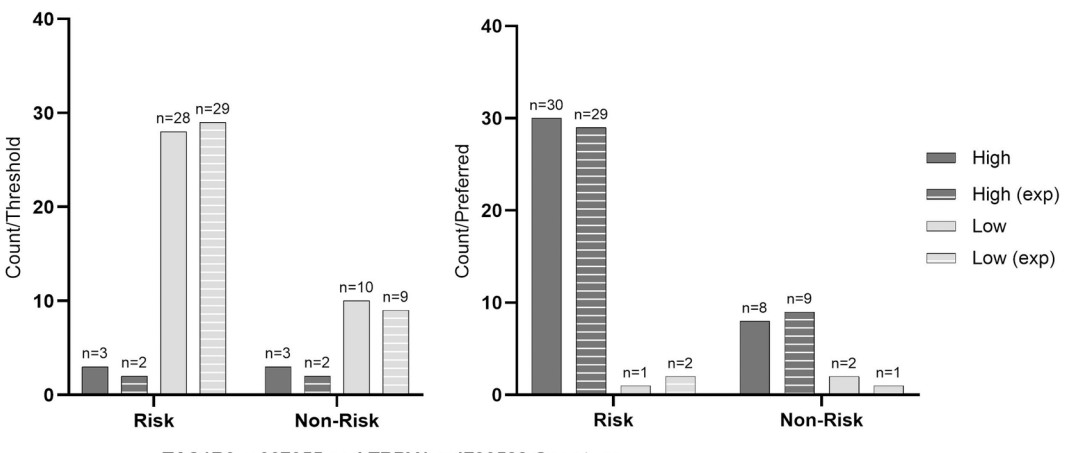

**Fig 3. Actual and expected frequency by combined genotypes of high and low threshold and preferred concentration groupings in healthy participants.** *Low Threshold; ≤ 15g/L, High Threshold; > 15g/L, Low Preferred Concentration; ≤ 30g/L, High Threshold; > 30g/L, TAS1R3; Taste 1 Receptor Member 3 gene, rs307355, TRPV1; Transient Receptor Potential Cation Channel Subfamily V Member 1 gene, rs4790522, Combined Risk; homozygous minor allele and homozygous minor/heterozygous groups, Combined No Risk; homozygous wildtype and homozygous wildtype/heterozygous groups. P-value significance at 0.05, no significance found. Figure represents Chi Squared contingency actual and expected counts between additive, recessive and dominant models.*

preference of sucrose was not investigated. It is important to acknowledge that social desirability bias [39] may have influenced the responses of the participants, due to data being collected in a hospital setting. Further research should endeavor to assess the relationship between PwT2D, taste threshold, preference and actual dietary intake.

Our study found no individual association between *TAS1R3* rs307355 or *TRPV1* rs4790522 and sweet taste threshold, preferred sample concentration nor any health metrics assessed. However, when combined genotypes were assessed a higher proportion of PwT2D with elevated preferred concentrations carried one or both homozygous risk alleles. This is supportive of polygenic risk factors influencing T2DM pathology [40,41]. A study conducted among a European population discovered that individual with alleles TC, SNPs located upstream of the TAS1R3 coding region, rs307355 and rs35744813, are highly related to human taste sensitivity to sucrose and individuals with T alleles are less susceptible to sucrose than people with C alleles [25]. A different study found that TAS1R3-linked polymorphic genotypes with heterozygous, TC, alleles at rs307355 are related to strong sweet taste sensitivity and that diabetes had a higher preference for sweetness [42]. Overall, highlighting the potential polygenic contributors to taste sensitivity. However, despite our novel cohort investigated, this study had a small sample size and thus combining genotypes was exploratory analysis, which could have influenced the results, therefore such results are hypothesis generating only. Additionally, given the paucity of knowledge regarding taste receptor genes in the African population it is possible other genetic variants, yet to be discovered, may contributed to a more accurate conclusion. Therefore, for results to be interpreted correctly further research on a larger sample size is required incorporating whole genome analysis

**Table 2. Participant characteristics of people with Type 2 Diabetes Mellitus (mean±SD), in the total cohort and by *TAS1R3* rs307355 and *TRPV1* rs4790522 genotypes, in an additive, recessive and dominant manner.**

| TAS1R3 | Total | †TT | CT | *CC | p-value | *TT/CT | p-value | †CT/CC | p-value |
|---|---|---|---|---|---|---|---|---|---|
| No. | 47 | 16 | 16 | 15 | | 32 | | 31 | |
| Age (yrs) | 55.9±13.0 | | | | | | | | |
| Sex (%F) | 68% | 75% | 75% | 53% | | 75% | | 65% | |
| BMI (kg/m²) | 28.0±5.5 | 27.7±5.0 | 28.0±5.8 | 28.2±6.1 | 0.975 | 27.9±5.3 | 0.866 | 28.1±5.9 | 0.829 |
| SBP (mmHg) | 135.0±22.1 | 128.8±20.8 | 134.3±21.9 | 142.3±22.8 | 0.232 | 131.5±21.2 | 0.119 | 138.2±22.3 | 0.167 |
| DBP (mmHg) | 80.3±12.2 | 76.9±14.3 | 81.1±9.9 | 83.1±12.1 | 0.355 | 79.0±12.3 | 0.282 | 82.1±10.9 | 0.171 |
| Pulse (bpm) | 85.9±12.7 | 81.4±11.6 | 87.3±11.2 | 89.0±14.6 | 0.271 | 84.4±11.6 | 0.284 | 88.1±12.8 | 0.086 |
| Glucose (mmol) | 9.4±4.7 | 10.5±7.3 | 8.6±2.2 | 9.1±3.1 | 0.817 | 9.6±5.4 | 0.664 | 8.8±2.6 | 0.536 |
| TRPV1 | Total | †CC | AC | *AA | p-value | *AC/CC | p-value | †AA/AC | p-value |
| | 47 | 12 | 21 | 14 | | 35 | | 33 | |
| Age (yrs) | 55.9±13.0 | | | | | | | | |
| Sex (%F) | 68% | 83% | 62% | 57% | | 30% | | 69% | |
| BMI (kg/m²) | 28.0±5.5 | 26.9±0.0 | 29.0±5.0 | 27.4±5.4 | 0.506 | 28.0±5.5 | 0.630 | 28.4±5.5 | 0.424 |
| SBP (mmHg) | 135.0±22.1 | 133.8±16.1 | 133.4±24.3 | 138.4±24.0 | 0.797 | 135.0±22.1 | 0.500 | 135.4±22.1 | 0.838 |
| DBP (mmHg) | 80.3±12.2 | 80.3±9.7 | 77.4±13.7 | 84.6±11.3 | 0.237 | 80.3±12.2 | 0.114 | 80.3±12.2 | 0.988 |
| Pulse (bpm) | 85.9±12.7 | 90.3±12.5 | 82.4±13.4 | 87.2±11.0 | 0.211 | 85.9±12.7 | 0.636 | 84.3±12.7 | 0.166 |
| Glucose (mmol) | 9.4±4.7 | 10.3±6.5 | 8.8±4.5 | 9.6±3.2 | 0.473 | 9.4±4.7 | 0.499 | 9.1±4.7 | 0.494 |

BMI; body mass index, DBP; diastolic blood pressure, Glucose; fasting blood glucose, No.; participant number, SBP, systolic blood pressure, TAS1R3; Taste 1 Receptor Member 3 gene, rs307355; TT (wildtype), CT, CC, TT/CT (recessive model), CT/CC (dominant model); TRPV1; Transient Receptor Potential Cation Channel Subfamily V Member 1 gene, rs4790522; CC (wildtype), AC, AA, AC/CC (recessive model), AC/CC (dominant model), *† indicate comparisons made within genotypes. P-value significance at 0.05, significant values are in bold. One way ANOVA, T-test, Kruskal Wallis, or Mann Whitney U tests were used where appropriate.

**Table 3. Participant characteristics of healthy participants and people with type 2 diabetes mellitus (mean±SD), in the total cohorts.**

| | Healthy | T2DM | p-value |
|---|---|---|---|
| No. | 42 | 47 | |
| Age (yrs) | 28.5±13.3 | 55.9±13.0 | **<0.001** |
| Sex (%F) | 50% | 48% | 0.064 |
| BMI (kg/m²) | 24.5±4.9 | 27.9±5.5 | **0.004** |
| SBP (mmHg) | 114.7±14.0 | 135.0±22.1 | **<0.001** |
| DBP (mmHg) | 70.3±10.1 | 80.3±12.2 | **<0.001** |
| Pulse (bpm) | 71.9±12.8 | 85.9±12.7 | **<0.001** |

BMI; body mass index, DBP; diastolic blood pressure, SBP, systolic blood pressure, T2DM; type 2 diabetes mellitus; P-value significance at 0.05, significant values are in bold. T-test, Mann Whitney U, Chi Squared or Fishers Exact used where appropriate.

Furthermore, other limitations were apparent, such as participant age range disparities. Taste sensitivity is known to decline with age [36] and one of the main risk factors for prediabetes and diabetes is advanced age. Due to our sample size limitations controlling for this factor was not possible, and thus comparative results (healthy versus PwT2D) warrant further result prior to applicability. It is imperative that further research considers this, particularly in Sub-Saharan African cohorts where research is so scarce. Additionally, measuring actual dietary, and specifically sugar, intake would have strengthened conclusions drawn, despite dietary intake measurements including errors [43–45]. it is important to explore whether differences in taste threshold and preferences are reflected in dietary intake. All considered, future research is required which is robustly powered to assess genetic variations, taste sensitivities, and actual dietary intake in a Sub-Saharan African cohort.

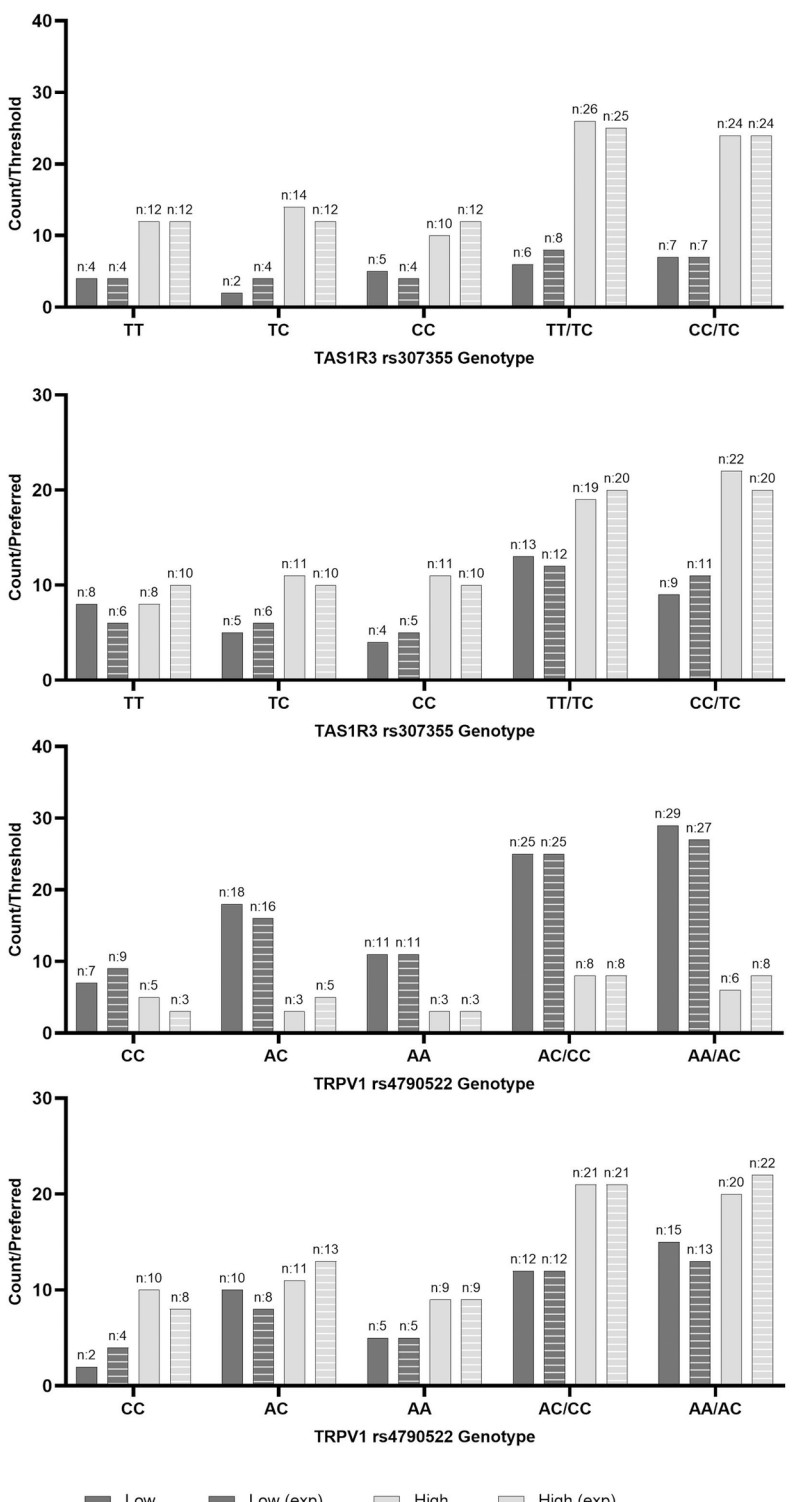

**Fig 4. Actual and expected frequency by genotype of high and low threshold and preferred concentration groupings in people with Type 2 Diabetes Mellitus.** *Low Threshold; ≤ 15g/L, High Threshold; >15g/L, Low Preferred Concentration; ≤ 30g/L, High Threshold; >30g/L, TAS1R3; Taste 1 Receptor Member 3 gene, rs307355; TT (wildtype), CT, CC, TT/CT (recessive model), CT/CC (dominant model); TRPV1; Transient Receptor Potential*

*Cation Channel Subfamily V Member 1 gene, rs4790522; CC (wildtype), AC, AA, AC/CC (recessive model), AC/CC (dominant model). P-value significance at 0.05, no significance found. Figure represents Chi Squared contingency actual and expected counts between additive, recessive and dominant models.*

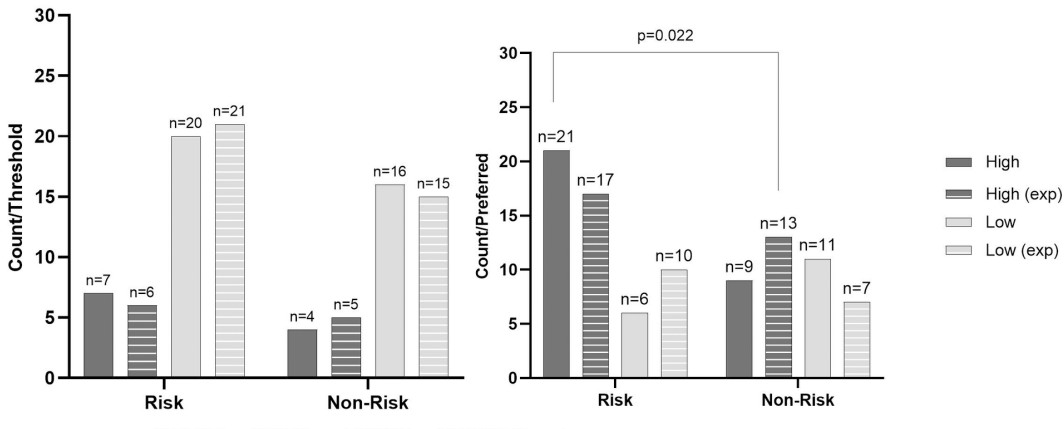

**Fig 5. Actual and expected frequency by combined genotypes of high and low threshold and preferred concentration groupings in people with Type 2 Diabetes Mellitus.** *Exp; expected value, Low Threshold; ≤ 15g/L, High Threshold; >15g/L, Low Preferred Concentration; ≤ 30g/L, High Threshold; >30g/L, TAS1R3; Taste 1 Receptor Member 3 gene, rs307355, Transient Receptor Potential Cation Channel Subfamily V Member 1 gene, rs4790522, Combined Risk; homozygous minor allele and homozygous minor/heterozygous groups, Combined No Risk; homozygous wildtype and homozygous wildtype/heterozygous groups; P-value significance at 0.05, significant values are provided. Figure represents Chi Squared contingency actual and expected counts between additive, recessive and dominant models.*

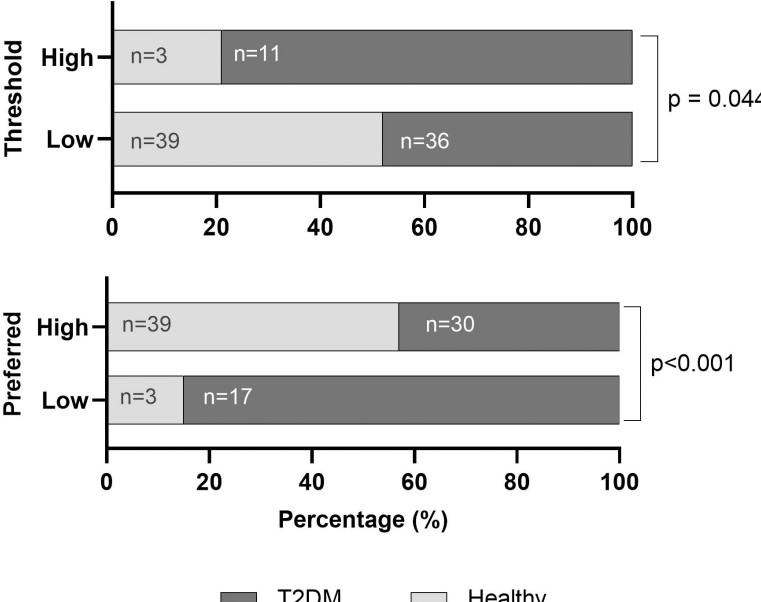

**Fig 6. Percentage differences between healthy participants and people with Type 2 Diabetes Mellitus and high and low threshold and preferred sample groupings.** Low Threshold; ≤ 15g/L, High Threshold; >15g/L, Low Preferred Concentration; ≤ 30g/L, High Threshold; >30g/L, T2DM; Type 2 Diabetes Mellitus; P-value significance at 0.05, significant values are provided. Chi Squared or Fishers Exact used where appropriate.

## Conclusions

This study provides novel insights into taste perception and genetic variation in a Sub-Saharan African population. Observed differences between PwT2D and healthy individuals highlights the potential role of altered taste perceptions and the importance of further research to inform tailored nutritional strategies aimed at improving glycemic control and metabolic outcomes. Such studies require a large sample size, with age matched participant groups. The preliminary evidence of genetic influencers on taste perception in this understudied population highlights a critical gap in data regarding taste sensitivity, receptor polymorphisms, and allele distribution in African cohorts. Therefore, further research is warranted including a wider array of taste-related SNPs, particularly those with higher frequencies in African populations. Overall, the findings contribute to closing this knowledge gap and support the need for larger, African population-specific studies to clarify genetic mechanisms and inform culturally relevant, personalized healthcare approaches.

## Supporting information

**S1 Strobe checklist.**
(XLSX)

**S2 Dataset.**
(DOCX)

**S1 Table 1.**
(DOCX)

**S2 Table 2.**
(DOCX)

## Acknowledgments

We are very grateful to all Laboratory personnel at St Mary's University, the Senior Medical Superintendent's Livingstone University Teaching Hospital (LUTH), Livingstone Center for Prevention and Translational Science and Mulungushi University for the support.

## Author contributions

**Conceptualization:** Tuku Mwakyoma, Leta Pilic, Sepiso K Masenga.

**Data curation:** Tuku Mwakyoma, Leta Pilic, Sepiso K Masenga.

**Formal analysis:** Tuku Mwakyoma, Catherine Anna-Marie Graham, Joreen P. Povia, Leta Pilic, Sepiso K Masenga.

**Funding acquisition:** Leta Pilic, Sepiso K Masenga.

**Investigation:** Tuku Mwakyoma.

**Methodology:** Tuku Mwakyoma, Catherine Anna-Marie Graham, Benson M Hamooya, Sepiso K Masenga.

**Resources:** Leta Pilic.

**Software:** Catherine Anna-Marie Graham.

**Supervision:** Leta Pilic, Sepiso K Masenga.

**Validation:** Tuku Mwakyoma, Catherine Anna-Marie Graham, Benson M Hamooya, Joreen P. Povia, Sepiso K Masenga.

**Visualization:** Tuku Mwakyoma, Catherine Anna-Marie Graham, Joreen P. Povia, Leta Pilic, Sepiso K Masenga.

**Writing – original draft:** Tuku Mwakyoma, Benson M Hamooya, Lweendo Muchaili, Memory Ngosa, Leta Pilic, Sepiso K Masenga.

**Writing – review & editing:** Tuku Mwakyoma, Catherine Anna-Marie Graham, Benson M Hamooya, Lweendo Muchaili, Memory Ngosa, Joreen P. Povia, Leta Pilic, Sepiso K Masenga.

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
