## [Decision Letter · Decision Letter 0]

Dear Dr. Mwakyoma,

Thank you for submitting your manuscript to PLOS ONE. After careful consideration, we feel that it has merit but does not fully meet PLOS ONE’s publication criteria as it currently stands. Therefore, we invite you to submit a revised version of the manuscript that addresses the points raised during the review process.

We look forward to receiving your revised manuscript.

Kind regards,

Academic Editor

PLOS ONE

Journal Requirements:

2. We note you have included a table to which you do not refer in the text of your manuscript. Please ensure that you refer to Table 2 in your text; if accepted, production will need this reference to link the reader to the Table.

Reviewers' comments:

Reviewer's Responses to Questions

**Comments to the Author**

1. Is the manuscript technically sound, and do the data support the conclusions?

Reviewer #1: Partly

Reviewer #2: Yes

2. Has the statistical analysis been performed appropriately and rigorously?

Reviewer #1: Yes

Reviewer #2: Yes

3. Have the authors made all data underlying the findings in their manuscript fully available?

Reviewer #1: Yes

Reviewer #2: Yes

4. Is the manuscript presented in an intelligible fashion and written in standard English?

Reviewer #1: Yes

Reviewer #2: Yes

Reviewer #1: The manuscript has low quality. It did not address any specific gap they intended to fill. Furthermore, it does not add anything to the subject area compared with other published Material. Therefore, any comments regarding the methodology are equivocal. In addition, the discussion was inappropriate. This manuscript does not fulfill the standards established for the journal to be considered for publication.

Reviewer #2: The Manuscript sounds ok according to the number of participants. I think sample size is small. But its ok but you should add more case studies to prove the data and add more references and improve little bit your english.

**Do you want your identity to be public for this peer review?** For information about this choice, including consent withdrawal, please see our Privacy Policy

Reviewer #1: **Yes: ** Shooka Mohammadi

Reviewer #2: **Yes: ** Sammra Maqsood

---

## [Author Response · Author response to Decision Letter 1]

10 Mar 2025

Response: Formatted as advised

2. We note you have included a table to which you do not refer in the text of your manuscript. Please ensure that you refer to Table 2 in your text; if accepted, production will need this reference to link the reader to the Table.

Response: The table has been cited

Response: Captions have been added for supporting information files

Comments to the Author

1. Is the manuscript technically sound, and do the data support the conclusions?

Reviewer #1: Partly

Reviewer #2: Yes

Response: we have made adjustments to ensure that the manuscript support the conclusions

2. Has the statistical analysis been performed appropriately and rigorously?

Reviewer #1: Yes

Reviewer #2: Yes

Response: Thank you

3. Have the authors made all data underlying the findings in their manuscript fully available?

Reviewer #1: Yes

Reviewer #2: Yes

Response: Thank you

4. Is the manuscript presented in an intelligible fashion and written in standard English?

Reviewer #1: Yes

Reviewer #2: Yes

Response: Thank you

5. Review Comments to the Author

Reviewer #1: The manuscript has low quality. It did not address any specific gap they intended to fill. Furthermore, it does not add anything to the subject area compared with other published Material. Therefore, any comments regarding the methodology are equivocal. In addition, the discussion was inappropriate. This manuscript does not fulfill the standards established for the journal to be considered for publication.

Response: We appreciate the reviewers comment. We have added more information to address the specific gap and have clearly written out the main goal of the study. Our study is unique in this population and we present novel results. One main reason for the apparent discrepancy with published material is because the black African population is genetically different so much that there are variations in SNPs. We think that our study will provide good baseline data for future studies that build on this field of study. Further, there are few studies which have investigated taste sensitivity and measured these genetic SNPs in African populations. We did a thorough search of literature as reported in the introduction and discussion to this effect and found no specific study.

We went ahead to add more SNPs to provide more relevance to the study. The limitation here is that the SNPs that may be associated with taste sensitivity in Caucasian populations where these studies are conducted may not necessarily be similar to our population. Measuring a wide ray of SNPs would be better in identifying SNPs common to African populations but the cost implications are prohibiting and beyond the scope of our study.

Reviewer #2: The Manuscript sounds ok according to the number of participants. I think sample size is small. But its ok but you should add more case studies to prove the data and add more references and improve little bit your English.

Response: We want to thank the reviewer for the suggestions. We have now added more single nucleotide polymorphisms to our data to enrich the manuscript and provided more clearer aims. We have also made editions to the grammar throughout to improve the manuscript. In addition, refer to our response to reviewer 1 for additional changes we have made to improve the manuscript

---

## [Decision Letter · Decision Letter 1]

Dear Dr. Mwakyoma,

Thank you for submitting your manuscript to PLOS ONE. After careful consideration, we feel that it has merit but does not fully meet PLOS ONE’s publication criteria as it currently stands. Therefore, we invite you to submit a revised version of the manuscript that addresses the points raised during the review process.

**ACADEMIC EDITOR: Please insert comments here and delete this placeholder text when finished.**

Indicate which changes you require for acceptance versus which changes you recommendAddress any conflicts between the reviews so that it's clear which advice the authors should followProvide specific feedback from your evaluation of the manuscript

publication criteria  and not, for example, on novelty or perceived impact.

We look forward to receiving your revised manuscript.

Kind regards,

Academic Editor

PLOS ONE

Journal Requirements:

Additional Editor Comments (if provided):

Reviewers' comments:

Reviewer's Responses to Questions

**Comments to the Author**

Reviewer #3: All comments have been addressed

Reviewer #4: (No Response)

2. Is the manuscript technically sound, and do the data support the conclusions?

Reviewer #3: Yes

Reviewer #4: Partly

3. Has the statistical analysis been performed appropriately and rigorously?

Reviewer #3: Yes

Reviewer #4: No

4. Have the authors made all data underlying the findings in their manuscript fully available?

Reviewer #3: Yes

Reviewer #4: Yes

5. Is the manuscript presented in an intelligible fashion and written in standard English?

Reviewer #3: Yes

Reviewer #4: Yes

Reviewer #3: The research question is interesting, and the methodology employed is generally appropriate. Additionally, the manuscript adheres to the general standards of scientific writing. However, there are some critical points that require attention to align with the publication standards of PLOS ONE.

Strengths

Relevance of the Study: The research tackles an important question in the field of genes which may influence sweet taste sensivity in people with T2DM, which contributes to advancing knowledge on this subject in african population.

Methodological Rigor: The study design is generally sound and detailed, facilitating reproducibility.

Clarity and Presentation: The manuscript is structured logically, with clear figures and tables that effectively complement the text.

Limitations

The authors mention many constraints, even thought you discussion of the limitations—such as potential biases, confounding variables, and the lack of significant results— enhance the transparency and rigor of the study.

Suggestions for Improvement

Addressing Sample Size Limitations: If feasible, increasing the sample size or discussing why the current size is appropriate given the study design would improve the manuscript.

Supplementary Analysis: Including additional analyses (such as dominant or recessive model in statiscal analysis, odds ratio) could help inccrease the robustness of the findings.

In general, it's a good article but does not fit Plos One scope as it is now.

Reviewer #4: Reviewer #1: Summary

Mwakyoma et al. investigated the relationship between genetic variations in TRPV1 (rs4790522, rs8065080) and TAS1R3 (rs307355) genes and sweet taste sensitivity in Zambian adults with and without type 2 diabetes mellitus (T2DM). Using a cross-sectional design (89 participants: 47 T2DM, 42 non-di2DM), they found reduced sweet taste sensitivity in T2DM individuals but no association between the studied SNPs and taste thresholds. The study highlights the need for larger cohorts and emphasizes potential non-genetic factors influencing taste perception in T2DM.

Overall Impression

The study addresses an under-researched topic in an African population, providing novel baseline data. However, limitations in sample size and methodology reduce the generalizability of conclusions.

The relatively small sample size in this study constrained the depth of subsequent analyses. However, given the availability of large-scale GWAS datasets from diverse ethnic populations (e.g., European and East Asian cohorts), the authors could further strengthen the findings through two complementary approaches:

1.Implementing Mendelian randomization (MR) analyses to investigate potential causal relationships between sweet taste sensitivity and T2DM across distinct ethnic groups;

2.Conducting cross-ancestry comparative analyses to evaluate whether the lead SNP effects/alleles associated with these traits are conserved between African populations and other ethnicities. Such trans-ethnic validation would enhance the generalizability of the findings while addressing population-specific genetic architectures.

**Do you want your identity to be public for this peer review?** For information about this choice, including consent withdrawal, please see our Privacy Policy

Reviewer #3: No

Reviewer #4: No

---

## [Author Response · Author response to Decision Letter 2]

30 Apr 2025

Dear editor,

Thank you for the communication and consideration of our manuscript. We have spent time to make improvements aligned with both reviewers’ comments and have responded in a point-counter-point manner below. We hope that you find the revised manuscript appropriate for publication.

Reviewer #3: The research question is interesting, and the methodology employed is generally appropriate. Additionally, the manuscript adheres to the general standards of scientific writing. However, there are some critical points that require attention to align with the publication standards of PLOS ONE.

We thank you for seeing value in our manuscript and hope that the revisions made align with the publication standard of PLOS ONE.

Limitations:

The authors mention many constraints, even thought you discussion of the limitations—such as potential biases, confounding variables, and the lack of significant results— enhance the transparency and rigor of the study.

Suggestions for Improvement:

Addressing Sample Size Limitations: If feasible, increasing the sample size or discussing why the current size is appropriate given the study design would improve the manuscript.

Unfortunately, we are unable to increase our sample size, this is due to funding restraints. We hope that the publication of our manuscript will warrant further interest in researching our cohort and encourage future studies, including our own, to achieve a larger sample size. We have referenced our sample size limitations throughout the discussion from line.

Supplementary Analysis: Including additional analyses (such as dominant or recessive model in statistical analysis, odds ratio) could help increase the robustness of the findings.

We have now included additive, dominant and recessive models throughout the manuscript. We also have included a genotype grouping related to those carrying homozygous alternate alleles of the genes studied. We believe this suggested improvement has strengthened our manuscript so we would like to thank you for the consideration. In addition, we have assessed the sample size limitations we have when working with such a novel cohort, and as a result have decided to remove the TRPV1 rs8065080 SNP as the minor allele is not represented enough in our cohort. We believe this makes our reported findings stronger for the remaining two genotypes.

The manuscripts results and conclusion have been extensively revise upon your comment, so we hope that you find this version suitable for publication.

Reviewer #4: Reviewer #1: Summary:

Mwakyoma et al. investigated the relationship between genetic variations in TRPV1 (rs4790522, rs8065080) and TAS1R3 (rs307355) genes and sweet taste sensitivity in Zambian adults with and without type 2 diabetes mellitus (T2DM). Using a cross-sectional design (89 participants: 47 T2DM, 42 non-di2DM), they found reduced sweet taste sensitivity in T2DM individuals but no association between the studied SNPs and taste thresholds. The study highlights the need for larger cohorts and emphasizes potential non-genetic factors influencing taste perception in T2DM.

We thank you for seeing value in our manuscript and agree regarding the necessity to encourage further research in African cohorts.

Overall Impression

The study addresses an under-researched topic in an African population, providing novel baseline data. However, limitations in sample size and methodology reduce the generalizability of conclusions. The relatively small sample size in this study constrained the depth of subsequent analyses. However, given the availability of large-scale GWAS datasets from diverse ethnic populations (e.g., European and East Asian cohorts), the authors could further strengthen the findings through two complementary approaches:

1.Implementing Mendelian randomization (MR) analyses to investigate potential causal relationships between sweet taste sensitivity and T2DM across distinct ethnic groups;

2.Conducting cross-ancestry comparative analyses to evaluate whether the lead SNP effects/alleles associated with these traits are conserved between African populations and other ethnicities. Such trans-ethnic validation would enhance the generalizability of the findings while addressing population-specific genetic architectures.

We thank you for providing some methodological alternatives and agree that we are working with a small sample size. However, unfortunately we are unable to increase our sample size nor utilise other databases for this current study. This is due to funding restrains (including personnel). However, we have adapted our approach based on an alternative reviewer’s suggestions, by incorporating an additive, dominant and recessive structure to our results section. We have also excluded one TRPV1 SNP due to poor representation of the alternate genotype. Our work seeks to explore group differences and associations only at this point, with the hope that publication will allow exposure of this under-research cohort leading to future, larger studies.

The manuscripts results and conclusion have been extensively revised upon the comments from both reviewers, so we hope that you find this version suitable for publication.

---

## [Decision Letter · Decision Letter 2]

Dear Dr. Mwakyoma,

Thank you for submitting your manuscript to PLOS ONE. After careful consideration, we feel that it has merit but does not fully meet PLOS ONE’s publication criteria as it currently stands. Therefore, we invite you to submit a revised version of the manuscript that addresses the points raised during the review process.

We look forward to receiving your revised manuscript.

Kind regards,

Md. Asaduzzaman, Ph.D., M. Engg.

Academic Editor

PLOS ONE

**Journal Requirements:**

Reviewers' comments:

Reviewer's Responses to Questions

**Comments to the Author**

Reviewer #5: All comments have been addressed

Reviewer #6: All comments have been addressed

2. Is the manuscript technically sound, and do the data support the conclusions?

Reviewer #5: Yes

Reviewer #6: Yes

3. Has the statistical analysis been performed appropriately and rigorously?

Reviewer #5: Yes

Reviewer #6: Yes

4. Have the authors made all data underlying the findings in their manuscript fully available?

Reviewer #5: Yes

Reviewer #6: Yes

5. Is the manuscript presented in an intelligible fashion and written in standard English?

Reviewer #5: Yes

Reviewer #6: Yes

**Reviewer #5: ** 1. Addressing Age Confounding (Most Critical):

o In the Results/Discussion: More explicitly discuss the potential impact of the age difference on the observed taste sensitivity differences. Can you quantify how much taste sensitivity typically changes with age based on existing literature to contextualize the findings?

o Statistical (if feasible): Even if not for primary analysis, an exploratory age-adjusted analysis (such as ANCOVA for taste threshold with group as factor and age as covariate, or stratifying by age if numbers permit) could provide some insight. If not feasible, state clearly why and reiterate this as a major caveat when interpreting the taste sensitivity difference between groups.

2. Combined Genotype Finding (PwT2DM Preference):

o Clearly state the number of individuals in each subgroup for this specific analysis (those with high/low preferred concentration carrying the combined risk/non-risk alleles).

o Emphasize its exploratory nature due to the small sample size and potential for it being a chance finding, especially in the context of multiple implicit comparisons. Label it clearly as hypothesis-generating.

3. Discussion of "No Genetic Link":

o Continue to emphasize that the "no genetic link" refers specifically to the two SNPs studied in this cohort and under this study's power limitations. It does not rule out the involvement of these genes through other variants, gene-gene/gene-environment interactions, or the involvement of other unstudied taste-related genes.

4. Clarity in Figures/Tables:

o Ensure all figures and tables clearly state the 'n' for each group/subgroup being compared, especially for genotype-based analyses. S1 Table 1 (combined genotypes for healthy) and the similar analysis for PwT2DM (Table 3 combined analysis leading to Fig 5) should clearly show participant numbers in each combined genotype category.

5. Future Directions (already good, but could be slightly expanded):

o Reiterate the need for larger, ideally age-matched, cohort studies in diverse African populations.

o Suggest investigating a wider panel of taste-related genes or using hypothesis-free approaches (e.g., GWAS, if feasible in the future for such cohorts) to identify novel variants associated with taste perception and T2DM risk in African populations.

o The paradox of lower sensitivity but lower preference in PwT2DM is intriguing. Future studies could explore this with detailed dietary records, psychological assessments of food reward/craving, and awareness of dietary recommendations.

**Reviewer #6:**  To the author,

Please correct this information:

1- The paragraph : Collection of saliva samples and single nucleotide polymorphism (SNP) genotyping,

Please correct the dominant and recessive alleles for the TRPV1, as both the recessive allele and dominant allele are AC/CC, which is written as the following:

TRPV1; Transient Receptor Potential Cation Channel Subfamily V Member 1 gene, rs4790522; CC (wild type), AC, AA, AC/CC (recessive model), AC/CC (dominant model);

2- The reference 33,

Pilic L, Mavrommatis Y. Genetic predisposition to salt-sensitive normotension 237 and its effects on salt taste perception and intake. Br J Nutr. 2018;120:721–238 731. doi:10.1017/S0007114518002027

Is investigating the salty taste and the SNPs in the SLC4A5 (rs7571842, rs10177833), SCNN1B (rs239345)and TRPV1 (rs8065080) genes

The primers were not designed but used according to previous studies.

Except the TRPV1 (rs8065080), it does cover the primers and the technique used for the study of TAS1R3 rs307355, and TRPV1 rs4790522

**Do you want your identity to be public for this peer review?** For information about this choice, including consent withdrawal, please see our Privacy Policy

Reviewer #5: No

Reviewer #6: **Yes: ** Luma Hassan Alwan Al Obaidy

---

## [Author Response · Author response to Decision Letter 3]

13 Jun 2025

Dear Editor,

Thank you again for the communication. We have acknowledged both reviewers’ comments and have responded in a point-counter-point manner below. We hope that you find the revised manuscript appropriate for publication.

Reviewer #5:

We thank you for your careful overview of our manuscript. We have considered all comments are responded below.

1. Addressing Age Confounding (Most Critical):

a. In the Results/Discussion: More explicitly discuss the potential impact of the age difference on the observed taste sensitivity differences. Can you quantify how much taste sensitivity typically changes with age based on existing literature to contextualize the findings?

b. Statistical (if feasible): Even if not for primary analysis, an exploratory age-adjusted analysis (such as ANCOVA for taste threshold with group as factor and age as covariate or stratifying by age if numbers permit) could provide some insight. If not feasible, state clearly why and reiterate this as a major caveat when interpreting the taste sensitivity difference between groups.

a. Thank you for your comments. We have extended the paragraph in the discussion (paragraph 2; line 355-358) directly related to this result. We hope this provides more clarity. We believe we previously acknowledged the difference in age as a limitation in the discussion paragraph 1 and 6. We have therefore left these sections unchanged.

b. ANCOVA or similar analysis with a covariate is not appropriate with the sample size of this study. We previously acknowledged this in the limitations section (line 397-398). Upon your comment, we have extended our original sentence to highlight this further (line 398-399.

2. Combined Genotype Finding (PwT2DM Preference):

a. Clearly state the number of individuals in each subgroup for this specific analysis (those with high/low preferred concentration carrying the combined risk/non-risk alleles).

b. Emphasize its exploratory nature due to the small sample size and potential for it being a chance finding, especially in the context of multiple implicit comparisons. Label it clearly as hypothesis-generating.

a. Thank you for this comment, we believe it increases the quality of our manuscript. We have provided sample size in text (line: 282-283) and in Figures 3 and 5. New figures have been resubmitted with the manuscript.

b. The section in the results section that discusses the combined genetic results (line 388-395) highlights the limitations you have discussed in this feedback. We have extended the section with phrases recommended, for further emphasis.

3. Discussion of "No Genetic Link":

Continue to emphasize that the "no genetic link" refers specifically to the two SNPs studied in this cohort and under this study's power limitations. It does not rule out the involvement of these genes through other variants, gene-gene/gene-environment interactions, or the involvement of other unstudied taste-related genes.

After an in-depth discussion with the authors, we have concluded that line 85-102 and 376-395 emphasise than many studies have shown candidate and polygenic genetic associations with taste. Thus, we have not amended the introduction or discussion. However, we thank you for indicating this phrasing in the abstract and have updated the results and conclusion accordingly (line: 42-53).

4. Clarity in Figures/Tables:

Ensure all figures and tables clearly state the 'n' for each group/subgroup being compared, especially for genotype-based analyses. S1 Table 1 (combined genotypes for healthy) and the similar analysis for PwT2DM (Table 3 combined analysis leading to Fig 5) should clearly show participant numbers in each combined genotype category.

Supplementary Table 1 already includes the sample size for genotype groupings, if we have misunderstood your comment please advise. As per feedback comment 2 we have updated Figure 3 & 5 accordingly. We have also updated Figures 2,3 and 6 accordingly.

5. Future Directions (already good but could be slightly expanded):

a. Reiterate the need for larger, ideally age-matched, cohort studies in diverse African populations.

b. Suggest investigating a wider panel of taste-related genes or using hypothesis-free approaches (e.g., GWAS, if feasible in the future for such cohorts) to identify novel variants associated with taste perception and T2DM risk in African populations.

c. The paradox of lower sensitivity but lower preference in PwT2DM is intriguing. Future studies could explore this with detailed dietary records, psychological assessments of food reward/craving, and awareness of dietary recommendations.

Thank you for the kind words related to our final section. We have discussed you feedback thoroughly and amended the conclusion section, specifically related to point a and b. We hope that you consider this expansion ready for publication.

Reviewer #6:

We thank you for your careful overview of our manuscript. We have considered all comments are responded below.

Please correct this information: 1- The paragraph: Collection of saliva samples and single nucleotide polymorphism (SNP) genotyping. Please correct the dominant and recessive alleles for the TRPV1, as both the recessive allele and dominant allele are AC/CC, which is written as the following: TRPV1; Transient Receptor Potential Cation Channel Subfamily V Member 1 gene, rs4790522; CC (wild type), AC, AA, AC/CC (recessive model), AC/CC (dominant model).

Thank you for spotting this typo, it has been corrected (line 192).

2- The reference 33: Pilic L, Mavrommatis Y. Genetic predisposition to salt-sensitive normotension 237 and its effects on salt taste perception and intake. Br J Nutr. 2018; 120:721–238 731. doi:10.1017/S0007114518002027, Is investigating the salty taste and the SNPs in the SLC4A5 (rs7571842, rs10177833), SCNN1B (rs239345) and TRPV1 (rs8065080) genes.

• The primers were not designed but used according to previous studies.

• Except the TRPV1 (rs8065080), it does cover the primers and the technique used for the study of TAS1R3 rs307355, and TRPV1 rs4790522

Reference 33 is related to the genotyping methodology (extraction, quantification, genotyping), not the assay/primer design. We have revised the section titled “Collection of saliva samples and single nucleotide polymorphism (SNP) genotyping”, which we hope better portrays this (line 158-181).

Kind regards,

The authors

---

## [Decision Letter · Decision Letter 3]

Gene variations and sweet taste sensitivity in Zambian adults with and without type 2 diabetes mellitus.

PONE-D-24-49147R3

Dear Dr. Mwakyoma,

We’re pleased to inform you that your manuscript has been judged scientifically suitable for publication and will be formally accepted for publication once it meets all outstanding technical requirements.

Kind regards,

Md. Asaduzzaman, Ph.D., M. Engg.

Academic Editor

PLOS ONE

Additional Editor Comments (optional):

Reviewers' comments:

Reviewer's Responses to Questions

**Comments to the Author**

Reviewer #5: All comments have been addressed

Reviewer #6: All comments have been addressed

2. Is the manuscript technically sound, and do the data support the conclusions?

Reviewer #5: Yes

Reviewer #6: Yes

3. Has the statistical analysis been performed appropriately and rigorously?

Reviewer #5: Yes

Reviewer #6: Yes

4. Have the authors made all data underlying the findings in their manuscript fully available?

Reviewer #5: Yes

Reviewer #6: Yes

5. Is the manuscript presented in an intelligible fashion and written in standard English?

Reviewer #5: Yes

Reviewer #6: Yes

Reviewer #5: Thank you for your edits. A follow up manuscript of more patients will be helpful. Additionally, research on African cohort will be helpful to the community.

Reviewer #6: To the Author,

The manuscripts entitled (Gene variations and sweet taste sensitivity in Zambian adults with and without type 2

diabetes mellitus) may shed light on the medical issues associated with diabetic patients.

Thanks for correcting the manuscript according to the comments.

**Do you want your identity to be public for this peer review?** For information about this choice, including consent withdrawal, please see our Privacy Policy

Reviewer #5: No

Reviewer #6: **Yes: ** Luma Hassan Alwan Al Obaidy

---

## [Editor Report · Acceptance letter]

PONE-D-24-49147R3

PLOS ONE

Dear Dr. Mwakyoma,

I'm pleased to inform you that your manuscript has been deemed suitable for publication in PLOS ONE. Congratulations! Your manuscript is now being handed over to our production team.

Kind regards,

on behalf of

Dr. Md. Asaduzzaman

Academic Editor

PLOS ONE